# Maternal Selenium Deficiency in Mice Alters Offspring Glucose Metabolism and Thyroid Status in a Sexually Dimorphic Manner

**DOI:** 10.3390/nu12010267

**Published:** 2020-01-20

**Authors:** Pierre Hofstee, Daniel R. McKeating, Lucy A. Bartho, Stephen T. Anderson, Anthony V. Perkins, James S. M. Cuffe

**Affiliations:** 1School of Medical Science, Menzies Health Institute Queensland, Griffith University Gold, Coast Campus, Southport, QLD 4215, Australia; p.hofstee@griffith.edu.au (P.H.); daniel.mckeating@griffithuni.edu.au (D.R.M.); lucy.bartho@griffithuni.edu.au (L.A.B.); a.perkins@griffith.edu.au (A.V.P.); 2The School of Biomedical Sciences, The University of Queensland, St Lucia, QLD 4072, Australia; stephen.anderson@uq.edu.au

**Keywords:** DOHaD, micronutrients, pregnancy, reproduction, selenoprotein, endocrine

## Abstract

Selenium is an essential micronutrient commonly deficient in human populations. Selenium deficiency increases the risks of pregnancy complications; however, the long-term impact of selenium deficiency on offspring disease remains unclear. This study investigates the effects of selenium deficiency during pregnancy on offspring metabolic function. Female C57BL/6 mice were allocated to control (>190 μg selenium/kg, *n* = 8) or low selenium (<50 μg selenium/kg, *n* = 8) diets prior to mating and throughout gestation. At postnatal day (PN) 170, mice underwent an intraperitoneal glucose tolerance test and were culled at PN180 for biochemical analysis. Mice exposed to selenium deficiency in utero had reduced fasting blood glucose but increased postprandial blood glucose concentrations. Male offspring from selenium-deficient litters had increased plasma insulin levels in conjunction with reduced plasma thyroxine (tetraiodothyronine or T4) concentrations. Conversely, females exposed to selenium deficiency in utero exhibited increased plasma thyroxine levels with no change in plasma insulin. This study demonstrates the importance of adequate selenium intake around pregnancy for offspring metabolic health. Given the increasing prevalence of metabolic disease, this study highlights the need for appropriate micronutrient intake during pregnancy to ensure a healthy start to life.

## 1. Introduction

It is well accepted that a healthy pregnancy is required to ensure optimal fetal development and a healthy start to life for the developing child. Deficiencies in key dietary components during pregnancy have been shown to increase offspring risk of developing conditions such as insulin resistance, type 2 diabetes mellitus (T2DM) and metabolic dysfunction [1]. Given that, globally, diabetes mellitus type 1 and T2DM combined now impacts approximately 1 in 11 adults [2], and rates of obesity and insulin resistance are increasing over time, understanding the factors that contribute to these diseases is of paramount importance. Several clinical studies have linked decreased micronutrient status to poor pregnancy outcomes [3]. Often clinical studies are confounded by deficiencies in macronutrients as well as multiple micronutrients, making it difficult to delineate exact mechanisms of action for individual perturbations [3]. In contrast, animal studies allow manipulation of single dietary components to assess how each factor may program disease outcomes in offspring.

Selenium is one micronutrient that is commonly deficient in the human diet, with both human and animal studies demonstrating that selenium deficiency can increase the risk of a range of pregnancy complications [4,5]. Human clinical studies have linked reduced maternal selenium status to an increased risk for pregnancy-induced hypotension [6], preeclampsia [7], gestational diabetes mellitus [8] and small-for-gestational age newborns [9]. Importantly, clinical studies have also demonstrated that pregnancy complications such as these are associated with adverse disease outcomes later in life [10,11]. However, the long-term impact of selenium deficiency during pregnancy on offspring disease outcomes is largely unknown. Selenium is required for the formation of 25 selenoproteins, many of which are important for regulating cellular homeostasis. The recommended dietary intake for selenium during pregnancy is 65 µg/day, with limited data supporting this recommendation [3,12]. Selenium deficiency impairs the function of selenoproteins, including glutathione peroxidases, which have been reported to influence regulation of enzymes in the insulin signalling cascade and glucose metabolism [13]. Additionally, selenoprotein N dysfunction is detrimental to myogenesis and skeletal muscle development [14]. As skeletal muscle is the largest site of glucose utilisation and is commonly implicated in the development of insulin resistance, it is possible that aberrant selenium levels may alter skeletal muscle development and contribute to impaired carbohydrate metabolism later in life [15].

Selenium is also known to play a major role in regulating thyroid hormone metabolism [16]. The selenoprotein iodothyronine deiodinases (DIO) regulate the production and metabolism of thyroid hormones. While DIO1 and DIO2 are localised to several peripheral tissues, DIO3 is expressed only within fetal tissues such as the placenta [17]. We have recently published data from a mouse model of selenium deficiency during pregnancy that demonstrates reduced placental expression of *DIO2* and *DIO3* and increased maternal and fetal thyroid hormone concentrations during pregnancy [18]. Thyroid hormones regulate numerous metabolic processes and are essential for healthy pregnancy outcomes and fetal development [19]. It is thus possible that selenium deficiency may program offspring disease indirectly through altering thyroid function. Our mouse model of selenium deficiency also resulted in placental insufficiency with reduced fetal glucose concentrations and fetal growth restriction, two factors also associated with programmed disease outcomes in offspring [18].

Given the disruption of fetal thyroid hormone and glucose concentrations that occurred during development, it is pertinent to investigate the long-term impact of selenium deficiency during pregnancy on thyroid hormone metabolism and glucose homeostasis in offspring. In this study, we aim to characterise the long-term effects of maternal dietary selenium deficiency prior to pregnancy, throughout gestation and during lactation on offspring metabolic health later in life. Given that programmed disease outcomes often occur in a sexually dimorphic manner, the potential biochemical pathways that regulate metabolic outcomes were assessed separately in male and female offspring.

## 2. Materials and Methods

### 2.1. Animal Procedures

All experiments were approved by the Griffith University Animal Ethics Committee, conducted in accordance with the Australian Code of Practice for Care and Use of Animals for Scientific Purposes, with all experimental protocols complying with policies and regulations approved by the Griffith University Animal Ethics Committee (MSC/01/16/AEC). Animal procedures, including mating and diet, have been previously described [18]. Housing and husbandry of animals, as well as conceptualisation of experimental design, was done so in accordance with the Developmental Origins of Health and Disease (DOHaD) research “Animals in Research: Reporting In Vivo Experiments” (ARRIVE) guidelines.

Briefly, female C57BL/6 mice were obtained from the Animal Resources Centre (ARC, Perth, Western Australia) and stored in environmentally controlled conditions of 23 °C and standard 12 h light/dark cycles, with additional environmental enrichment. After acclimatisation for one week, mice were randomly allocated to either a control (>190 µg Se/kg, *n* = 8) or low selenium (<50 µg Se/kg, *n* = 8) diet four weeks prior to mating, throughout gestation and lactation. The custom diet used in the model has been previously described [18]. Quantities of selenium within the diet were verified using inductively coupled plasma mass spectrometry. After mice gave birth, offspring were left unhandled until postnatal day (PN) 8 from which point in time they were weighed daily. Offspring were weaned at PN24 and after which, were placed on normal animal chow (230 µg selenium/kg, Teklad Global 18% Protein Rodent Diet Irradiated, ENVIGO, Madison, WI, USA). All mice had access to water and their respective diets ad libitum. From weaning (PN24) onwards, mice were group-housed with one or two other litter mates of the same sex, with food and water manually weighed daily to calculate food and water intake per cage. Daily food and water intake values were determined daily until PN180. Average food and water intake was determined by dividing by the number of animals per cage. One male and one female from each litter were culled at PN30 via cervical dislocation (*n* = 1 pup per sex from 6–8 litters per group) and plasma/tissues collected. The remainder of the litter was aged until PN180 and then also culled via cervical dislocation for tissue collection (*n* = 6–8).

### 2.2. Post-Mortem Tissue Collection

At PN30 and PN180, blood was collected via cardiac puncture into lithium heparin tubes (HEP; Microvette CB 300 µL, Lithium Heparin, SARSTEDT, Nümbrecht, Germany; Cat. no. 16.443), which were subsequently centrifuged at 2000× *g* for 5 min followed by collection of blood plasma. The liver, adrenals, skeletal muscle and reproductive organs were collected, wet weighed, snap-frozen in liquid nitrogen and stored at −80 °C.

### 2.3. Glucose Tolerance Testing

Glucose levels were measured using an Accu-Chek Performa II glucometer (Mannheim, Germany). Random blood glucose levels were measured in whole blood at the time of euthanasia at PN30 (*n* = 6–8) or PN180 (*n* = 6–8). Fasting blood glucose levels were measured at PN90 and PN170 (*n* = 6–8). This involved fasting mice overnight, followed by a tail snip the following morning at 10:00 a.m. to assess fasting blood glucose. Immediately following the fasting blood glucose measurement at PN170, mice were subjected to an intraperitoneal (IP) glucose tolerance test (GTT). Mice were administered with 1 g/kg of glucose (D-(+)-Glucose, Sigma-Aldrich, Darmstadt, Germany) in saline (0.9% sodium chloride solution, Sigma-Aldrich) via IP injection (20% Glu in 0.9% NaCl solution). Glucose levels were measured at 0-, 30-, 60-, 90-, 120- and 180-min post-injection.

### 2.4. Hormone Analysis

Levels of insulin, corticosterone, tetraiodothyronine (thyroxine) and triiodothyronine were measured in plasma at PN180 as previously described [18,20]. Briefly, insulin was measured in duplicate using a single assay with the Ultra-Sensitive Mouse Insulin ELISA kit (Crystal Chem Inc., Elk Grove Village, IL, USA; Cat. no. 90080) following the manufacturer’s guidelines (*n* = 5–7). Corticosterone levels were also measured in duplicate within a single assay using the DetectX Corticosterone Enzyme Immunoassay kit (Arbor Assays, Ann Arbor, MI, USA; Cat. no. K014-H1) following the manufacturer’s protocol (*n* = 5–7). Thyroxine was measured in offspring plasma using a competitive ELISA kit (Invitrogen, Carlsbad, CA, USA; Cat. no. EIAT4C). Triiodothyronine (T_3_) was measured using a total T_3_ radioimmunoassay (RIA) kit (Beckman Coulter, Czech Republic; Cat. no. IM1699). Concentrations of all hormones in plasma were measured at the same time with respective calibrators, with final concentrations determined from the standard curve developed using the provided protocols.

### 2.5. Quantitative PCR

RNA was extracted from the gastrocnemius using the RNeasy mini kit (Qiagen, Melbourne, Australia; Cat. no. 74106), as described previously [18,20]. As this was muscle tissue, an additional step was included, which involved the addition of proteinase K (20 mg/mL) and incubation at 55 °C for 1 h. The final RNA concentration was measured via absorbance spectrophotometry using the NanoDrop 2000/2000c. Conversion of RNA into cDNA for qPCR analysis was completed using the Bio-Rad iScript gDNA clear cDNA synthesis kit (Hercules, CA, USA; Cat. no. 172-5035). All PCRs were performed with thermocycling parameters as follows: initial activation step—95 °C for 2 min, followed by 40 cycles of denaturation—95 °C for 5 s and combined annealing/extension 60 °C for 10 s on the StepOne real-time PCR system (Applied Biosystems, Carlsbad, CA, USA). Twenty nanograms of cDNA in 10 µL per reaction were used to perform qPCR, with samples run in duplicate. All PCR reactions were performed in correspondence with the *MIQE* guidelines.

Measurement of mRNA expression of metabolic and thyroid pathway genes was conducted using KiCqStart SYBR green PCR primers (Sigma-Aldrich, St. Louis, MI, USA), as described in Table 1. No product was detected in the non-template control on any run and melt curve analysis demonstrated a single spike for all genes measured. All samples were run in triplicate within a single plate per gene. All expression was normalised to the geometric mean of hypoxanthine phosphoribosyl transferase 1 (*Hprt1*, NM_013556) and beta-2-microglobulin (*B2m*, NM_009735). Several potential reference genes were assessed; however, these genes demonstrated consistent expression in the skeletal muscle tissue and were not impacted by treatment. Final expression was calculated using the 2-ΔΔCt method, with *n* = 6–8 per group.

### 2.6. Western Immunoblotting

Total protein was extracted from approximately 60 mg of frozen gastrocnemius tissue, as previously described [20]. Quantitation of total protein was conducted using a bicinchoninic acid (BCA) protein assay, with all samples normalised to 2 µg/µL for use in Western blotting procedures. Protein samples (6 per group) were loaded into 15-well 12% polyacrylamide gels followed by separation via electrophoresis at 120 V for approximately 2 h. Following electrophoresis, total protein loading for each lane on the membrane was assessed using Revert^TM^ 700 Total Protein stain, as per the manufacturer’s instructions. After each membrane was scanned, the total protein stain was removed before membranes were blocked using Odyssey blocking buffer (Millipore, Burlington, MA, USA) for 1 h, then incubated with the following antibodies overnight: anti-glucose transporter 4 (GLUT4, 1:1000; Abcam, Cat. no. ab654) or protein kinase B (Pan-AKT, 1:1000; Cell Signalling, Cat. no. 4691S). Membranes were washed, followed by a one-hour incubation with Li-Cor secondary antibodies (1:10,000, IRDye 680 goat anti-rabbit). Finally, protein expression was visualised with the Li-Cor Odyssey CLX imaging system. Protein expression of GLUT4 and Pan-AKT was normalised to total protein content.

### 2.7. Statistical Analysis

A maximum of one male and one female was selected from each litter for each of the analyses at PN30 and PN180. As some litters had limited numbers of female offspring, that meant that for some litters, less than *n* = 8 was available for analysis (PN30 and PN180 *n* = 6–8). Weight at PN30 and PN180, food and water consumption, allometry, as well as glucose and hormone concentrations, were analysed by two-way analysis of variance (ANOVA—GraphPad Prism 8.2.1; RRID:SCR 002798). The main effects of maternal selenium deficiency (Treatment; *P*_trt_) and sex (Sex; *P*_sex_), with any interactions between treatment and sex, were assessed (Interaction; *P*_int_). When a major effect of sex, treatment or an interaction between sex and treatment was detected, a Sidak post hoc analysis was performed. Given that male samples were run on a separate plate for QPCR and a separate gel for Western blotting compared to females, analysis of qPCR and western blotting was conducted using unpaired *t*-tests. All data are presented as means ± SEM with *p* < 0.05 considered statistically significant.

## 3. Results

### 3.1. Offspring Weights, Food and Water Consumption

As the impact of maternal selenium deficiency on offspring development has not been previously investigated, it is important to monitor how this developmental insult impacts offspring weight and dietary behaviour. Overall, LS did not impact pup number per litter but a sex by treatment interaction (*P_int_* < 0.05) and posthoc analysis (*p* < 0.05) demonstrated a reduction in the number of female but not male offspring (NS Male 2.13 ± 0.30, NS Female 3.5 ± 0.57, LS Male 2.63 ± 0.46, LS Female 1.63 ± 0.57). Body weight at PN30 was not different between offspring from normal and selenium-deficient litters (Figure 1A). Female weight was significantly less than male weight at PN30 (*P*_sex_ < 0.05). These results were reflected at PN180 (Figure 1B—*P*_sex_ < 0.0001). Food and water consumption from weaning (PN24) until PN180 was also not different between treatment groups; however, females ate significantly less (*P*_sex_ < 0.01) and drank significantly less (*P*_sex_ < 0.001) than males, regardless of treatment, over a 6-month period.

### 3.2. PN30 Offspring Allometry

Key parameters were investigated at PN30, which is roughly equivalent to that of an adolescent human, a time that often precedes the development of pathologies associated with non-communicable diseases (Table 2). At PN30, mice from selenium-deficient litters had increased tibialis anterior (TA) muscle weight (*P*_trt_ < 0.05), with posthoc analysis demonstrating that selenium deficiency increased TA weight by 26% in male offspring (*p* < 0.05). While selenium deficiency had no overall impact on the weight of other skeletal muscles, a sex by treatment interaction demonstrated that the extensor digitorum longus (EDL) was 18% heavier in male offspring from selenium-deficient pregnancies compared to male offspring from control pregnancies (Table 2, *P*_int_ < 0.05, *p* < 0.05). Muscles from male offspring were heavier than female offspring (*P*_sex_ < 0.05)

### 3.3. PN180 Offspring Allometry

Physiological and metabolic parameters were investigated at approximately PN170–180, which in mice is equivalent to mature adulthood, an equivalent age at which metabolic dysfunction may become apparent in humans (Table 3). At PN180, there was no difference in the weight of any organs between mice from normal selenium and selenium-deficient litters. As expected, organ weights from male offspring were heavier than female offspring (*P*_sex_ < 0.0001), except for the adrenal which was heavier in females than males. The heavier orgnans in males than females reflects the lower body weights of females at PN180 observed in Figure 1B.

### 3.4. Glucose Metabolism

As we had previously demonstrated that selenium deficiency reduced blood glucose levels in the fetus [18], glucose metabolism was investigated in the offspring from this model. At PN30, blood glucose levels were significantly reduced in the low selenium group (*P*_trt_ < 0.01, Figure 2A), irrespective of sex. Posthoc analysis indicates this was equally reduced in both sexes (*p* < 0.05). At PN90 (Figure 2B), fasting blood glucose levels were not impacted by selenium deficiency but were significantly lower in females compared to males (*P*_sex_ < 0.05). An interaction between treatment and sex (*P*_int_ < 0.05) suggested that selenium deficiency may have impacted PN90 fasting blood glucose in one sex but not the other; however, posthoc analysis did not reach statistical significance. Fasting blood glucose concentrations at PN170 were not affected by perinatal selenium deficiency (Figure 2C), although females had a reduced fasting blood glucose (*P*_sex_ < 0.001), compared to males.

In contrast to the fasting plasma glucose concentrations, the postprandial glucose assessment at PN180 was shown to be impacted by perinatal selenium deficiency. The area under the glucose tolerance curve (AUC, Figure 2D) was increased in mice from selenium-deficient litters (*P*_trt_ < 0.01), more so in females (*p* < 0.01). Irrespective of treatment, the AUC was reduced in females compared to males (*P*_sex_ < 0.0001). The peak blood glucose (Figure 2E) occurred at approximately 30 min post IP glucose injection in all groups; however, offspring from selenium-deficient litters had significantly increased blood glucose values at this peak (*P*_trt_ < 0.01) with posthoc analysis demonstrating this to be occurring predominantly in females (*p* < 0.01). Approximately three hours post IP injection of glucose (Figure 2F), blood glucose was still significantly increased in mice from selenium-deficient litters (*P*_trt_ < 0.05), again more so in females (*p* < 0.05).

### 3.5. Endocrine Analysis

Each of the major hormones that regulate glucose homeostasis were investigated. Plasma insulin levels at PN180 (Figure 3A) were significantly greater in males compared to females (*P*_sex_ < 0.001). While overall, selenium deficiency did not impact offspring insulin concentrations (*P*_trt_ = 0.2), a sex by treatment interaction (*P*_int_ < 0.05) and posthoc analysis (*p* < 0.05) demonstrated that male offspring of selenium-deficient mothers had a 54% increase in plasma insulin concentrations compared to controls. Although plasma corticosterone levels (Figure 3B) were greater in females than males (*P*_sex_ < 0.01), there were no differences due to treatment in either sex. There were significant differences in thyroxine levels (Figure 3C) between males and females (*P*_sex_ < 0.0001) and a significant interaction between sex and treatment (*P*_int_ < 0.01). Posthoc analysis determined that thyroxine levels were reduced in males (*p* < 0.01) and increased in females (*p* < 0.05) from dams exposed to selenium-deficient diets. T_3_ concentrations were not impacted by sex or treatment (Figure 3D).

### 3.6. Skeletal Muscle Metabolic Protein and Gene Expression

Given the major role of skeletal muscle in regulating systemic carbohydrate metabolism and insulin signalling, key genes and proteins implicated in insulin resistance were investigated (Figure 4). Males from selenium-deficient litters had significantly reduced expression of the insulin-regulated glucose transporter, GLUT4 (*Slc2a4*, Figure 4A—*p* < 0.01) and the insulin receptor (*Insr*, Figure 4B—*p* < 0.05). In addition, male offspring of selenium-deficient dams had reduced expression of key insulin signalling genes: phosphatidylinositol 3-kinase catalytic subunit Type 3 (*Pik3c3*, Figure 4C—*p* < 0.01), protein kinase B2 (*Akt2*, Figure 4D—*p* < 0.05) and phosphoglycerate kinase 1 (*Pgk1*, Figure 4F—*p* < 0.05). Females from selenium-deficient litters had increased expression of *Akt2* (Figure 4D—*p* < 0.05) within the gastrocnemius; however, no other genes were affected. Investigations into the protein expression of GLUT4 and Pan-AKT in the gastrocnemius of both males and females were also conducted. Protein expression of GLUT4 was reduced in deficient male offspring of selenium-deficient dams (*p* < 0.05, Figure 4G). Conversely, protein expression of GLUT4 was increased (*p* < 0.05) in female offspring of selenium-deficient dams compared to controls. Pan-AKT protein expression levels were reduced by approximately 30% in skeletal muscle of male offspring (*p* < 0.05, Figure 4H).

### 3.7. Skeletal Muscle Thyroid Hormone Gene Expression

As plasma thyroxine levels were dysregulated in a sex-specific manner at PN180, expression of genes involved in thyroid hormone signalling was also investigated in skeletal muscle (Figure 5). Thyroid hormone receptor α (Thr1, Figure 5A), the thyroid hormone transporter monocarboxylate transporter 10 (*Mct10*, Figure 5C) and uncoupling protein 3 (Ucp3, Figure 5D) were all significantly reduced (*p* < 0.05) in males from selenium-deficient litters. Conversely, expression of *Mct10* (Figure 5C) and Ucp3 (Figure 5D) were increased within the gastrocnemius of female offspring from selenium-deficient litters (*p* < 0.05), with no changes in Thra or Thrb gene expression.

## 4. Discussion

The effects of poor micronutrient status during pregnancy on DOHaD is of growing interest, and elucidating the implications on adult physiology is of paramount importance. The specific effects of individual deficits of essential trace elements such as selenium on pregnancy outcomes, placental function and fetal development are becoming increasingly known [3,21]; however, the specific consequences of maternal selenium deficiency on offspring metabolic physiology have not previously been demonstrated. We have recently shown that maternal selenium deficiency during pregnancy increases maternal and fetal thyroid hormone concentrations, alters fetal and placental glucose metabolism and induces fetal growth restriction [18]. As selenium deficiency induced variances to these systems during pregnancy, we hypothesised that alterations to glucose metabolism and thyroid hormone homeostasis may persist for offspring into adult life [22]. In this study, we examined the effects of maternal selenium deficiency during pregnancy on offspring metabolism and endocrine status. Given the major role of skeletal muscle tissue for systemic glucose homeostasis, we also investigated if any programmed metabolic dysfunction in offspring may be caused by changes to key insulin and thyroid signalling pathways in muscle tissue.

A major finding of the current study is that offspring from selenium-deficient pregnancies developed glucose intolerance by PN180. This is the first study to demonstrate that a mild selenium deficit during pregnancy may program alterations in the metabolic state of offspring. It is likely that this alteration in offspring glucose metabolism may be a consequence of altered glucose concentrations during fetal life [23]. Our previous publication demonstrated that selenium deficiency reduces fetal blood glucose levels at E18.5 [18]. As glucose is the primary energy substrate for the developing fetus [24], this likely contributed to the fetal growth restriction evident in this model. In the current study, we show that blood glucose levels remain reduced at PN30 in offspring of selenium-deficient dams, although fasting glucose concentrations were no longer different between treatment groups by PN90. Whilst fasting blood glucose remained unaffected by prenatal diet at PN180, postprandial glucose concentrations were elevated and remained elevated three hours after glucose administration in selenium deficient offspring. Given that the fetus completed its gestation in an environment with reduced glucose availability, the “thrifty phenotype” hypothesis suggests that the fetus would have adapted to a low glucose environment by altering mechanisms to increase glucose availability to ensure survival [25]. It is therefore unsurprising that once the offspring began to consume adequate nutrients in the post-weaning environment, the capacity to handle glucose was perturbed.

Plasma levels of insulin were also significantly increased in male offspring of selenium-deficient dams at PN180, which is suggestive of insulin resistance [26]. Given that clinical studies have shown selenium deficiency can reduce birth weight [9], a known predictor of offspring insulin resistance, this may be the mechanism through which this occurs. Investigations into the Dutch famine (1944–1945) have shown that when women were impacted by severe macronutrient deficiency during pregnancy, offspring were similarly born with a low birth weight with adult offspring developing impaired glucose tolerance [27]. Previous studies have demonstrated that programmed glucose intolerance and insulin resistance may be due to alterations to skeletal muscle physiology [28]. Ozanne et al. demonstrated that perinatal protein restriction in rats results in reduced skeletal muscle mass and while metabolic health appeared to be improved during adolescence, when the animals were aged into adulthood, insulin sensitivity and glucose uptake were significantly reduced in skeletal muscle [29]. In the current study, skeletal muscle size was increased in the adolescent offspring; however, at PN180, skeletal muscle size was similar between treatments. Furthermore, at E18.5, it was shown that placental selenoprotein N (SelN) expression was reduced [18]. During embryogenesis, SelN is important for muscle organisation and myoblast proliferation [30]. SelN may have been similarly impaired in the developing skeletal muscle, which could have altered muscle formation possibly contributing the phenotype observed in adulthood [31]. It is likely that the programmed phenotype in offspring has occurred because of complex interactions between reduced fetal glucose, altered skeletal muscle formation as well as changes to circulating hormones.

The manifestation of insulin resistance in skeletal muscle occurs due to reduced insulin-stimulated glucose uptake as well as impaired post-receptor and intracellular signalling pathways [26]. At PN180, mRNA and protein expression of the GLUT4 receptor was reduced in males and increased in females exposed to a low-selenium diet during early development. Interestingly, expression of *Akt2* was decreased in males and increased in females and at the protein level, Pan-AKT was reduced in males. Furthermore, mRNA expression of several genes involved in the insulin receptor pathway and glucose metabolism were also decreased in male skeletal muscle at PN180, including *Insr, Pik3c3 and Pgk1*. Abdul-Ghani and DeFronzo described insulin receptor alterations, decreased PI3-kinase activation, impaired GLUT4 translocation and reduced glucose phosphorylation in the pathogenesis of skeletal muscle insulin resistance [26]. Similarly, these observations have been associated with programmed metabolic disease and the pathogenesis of insulin resistance [32]. Female offspring in the same model also have glucose intolerance; however, they exhibit increased GLUT4 protein expression and intracellular mRNA expression of *Akt2*, indicating attempts of positive adaptations to glucose intolerance [33]. These results indicate exposure to selenium deficiency in utero causes a sexually dimorphic adaptation of glucose metabolism in offspring. While we have previously demonstrated that sexually dimorphic placental adaptations to maternal dietary insults likely mediate sex-specific disease outcomes in offspring, the placental adaptations noted in the current model were mostly similar between sexes [3]. The sexually dimorphic metabolic phenotype in the current study therefore likely reflects postnatal differences between males and females. While there are likely to be multiple postnatal factors involved [34], it would be of significant interest to investigate the potential role of estrogen in protecting female offspring from programmed metabolic disease in this model. This would be of particular interest given that increased concentrations of estrogen in females have been previously shown to play a major role in sexual dimorphic counterregulatory responses in healthy humans exposed to an acute bout of hypoglycaemia [35]. Another key hormone known to be different between men and women that is an important regulator of glucose homeostasis is cortisol. In the current model, while prenatal treatment had no impact on adrenal size or corticosterone concentrations, females had larger adrenal glands and increased plasma corticosterone levels when compared to males. Corticosterone has been previously shown to inhibit insulin-stimulated glucose uptake in skeletal muscle [36] and so it may be possible that the increased corticosterone concentrations in female offspring may play a role in modulating the programmed phenotype in the current study.

In addition to the potential roles of fetal hypoglycaemia and altered skeletal muscle function in the development of offspring disease, selenium deficiency may have altered offspring metabolic function as a consequence of programming thyroid dysfunction [37]. Korevaar et al. have previously demonstrated that maternal thyroid function is associated with offspring brain morphology and thyroid function in children at six years of life [38]. Animal models have similarly demonstrated that altered maternal thyroid status can disrupt the setpoint of the hypothalamus-pituitary-thyroid (HPT) axis in offspring [39]. Previously, we demonstrated that maternal selenium deficiency reduces placental *DIO* expression and increases maternal and fetal thyroid hormone levels [18]. Here, we show that thyroid hormone status remains impaired in offspring at PN180. In male offspring from mothers exposed to a selenium-deficient diet, plasma thyroxine levels were reduced, with no changes in triiodothyronine levels. Expression of genes known to be stimulated by thyroid hormone signalling was also reduced in the male gastrocnemius. In humans, negative associations between free thyroxine and insulin sensitivity suggest a potential role for thyroid hormone signalling in the development of insulin resistance [40]. Specifically, subclinical hypothyroidism is associated with insulin resistance, as seen in the male offspring [41].

In contrast to the programmed decrease in thyroxine in male offspring, selenium deficiency programmed an increase in thyroxine in female offspring. Insulin resistance is also demonstrated to be associated with hyperthyroidism, due to an increased demand for glucose, as observed in the female offspring in this study [42]. Prevalence of thyroid disorders is commonly higher in females [43]; therefore, it is possible that the observed sexual dimorphism may be in part due to sexually specific adaptive processes of the thyroid gland to stress [44]. Sexual dimorphic programmed changes to the HPT axis have been observed previously in animal models that investigate developmental programming, with sex-specific responses of the HPT axis to a high-fat/high-carbohydrate diet previously being demonstrated in rats [45]. In support of the potential impact that increased thyroid hormone levels may play in female offspring, expression of genes known to be stimulated by thyroid signalling was increased in muscle tissue from females, specifically *Mct10* and *Ucp3*. Glucose intolerance observed in females was also more severe than that observed in male offspring, with no change in plasma insulin levels at PN180. As discussed above, the sexual dimorphic variances observed may be due to endocrine function differences between males and females.

The novel findings of this study that demonstrate prenatal selenium deficiency induces sexually dimorphic changes to offspring metabolic health, particularly those related to thyroid function, highlight the urgent need for further research to investigate the role of sex hormones in programmed thyroid dysfunction. Furthermore, given that previous studies have demonstrated the benefits of pre/postnatal exercise in the prevention of programmed disease, and that exercise can stimulate GLUT4 glucose uptake in skeletal muscle independent of insulin, future studies should investigate whether exercise can ameliorate some of the effects demonstrated in the current study [46,47].

## 5. Conclusions

The current study is the first to demonstrate that selenium deficiency, from before pregnancy up until weaning, can program metabolic and thyroid dysfunction in offspring. Importantly, we have identified programmed changes to insulin and thyroid signalling within the skeletal muscle as likely causes of the offspring disease. Given the large number of populations that are selenium-deficient and the increasing rates of thyroid dysfunction and diabetes mellitus, it is important that action is taken to ensure women obtain adequate selenium during pregnancy to maximise the health of the developing child.

## Figures and Tables

**Figure 1 nutrients-12-00267-f001:**
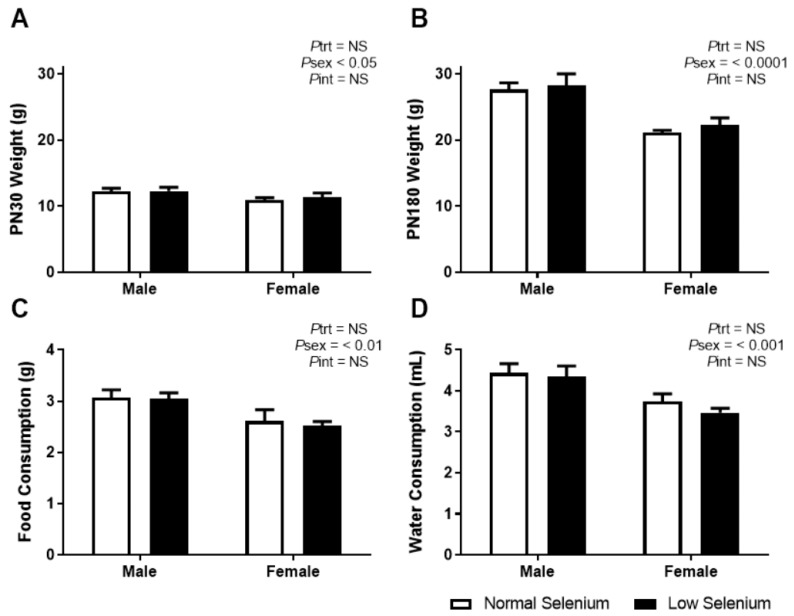
Offspring weights, food and water consumption. Body weight at (**A**) PN30 and (**B**) PN180 as well as average daily consumption of (**C**) normal chow and (**D**) water from weaning until PN180. Open bars (white) indicate offspring from litters that consumed a normal selenium diet, whereas closed bars (black) are offspring from litters that consumed a low selenium diet throughout pregnancy and up until weaning. Data are mean ± SEM and analysed by two-way ANOVA with treatment (*P*_trt_) and sex (*P*_sex_) as major factors. *P*_int_ represents the interaction between trt and sex. Multiple comparisons were determined by Sidak posthoc testing. Significance determined by *p* < 0.05. *n* = 6–9.

**Figure 2 nutrients-12-00267-f002:**
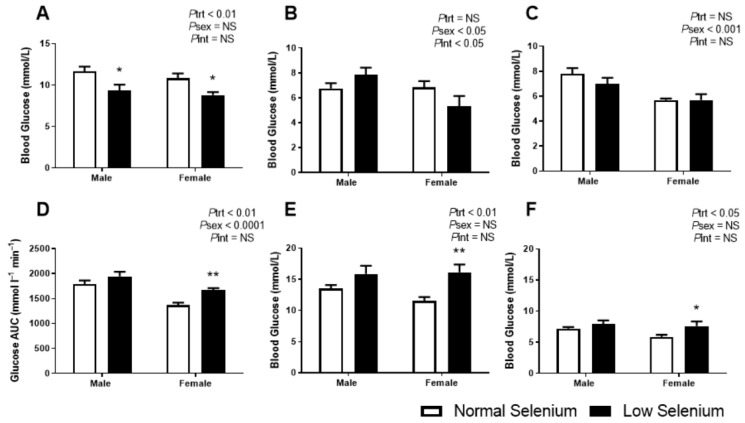
Offspring glucose metabolism. Offspring random glucose concentrations at (**A**) PN30 and fasting blood glucose concentrations at (**B**) PN90 and (**C**) PN170. Following an intraperitoneal (IP) of glucose at PN170, the following parameters were determined: (**D**) GTT area under the glucose curve (AUGC), (**E**) peak blood glucose concentrations 30 min post IP and (**F**) blood glucose concentrations 180 min post IP. Open bars (white) indicate offspring from litters that consumed a normal selenium diet, whereas closed bars (black) are offspring from litters that consumed a low selenium diet throughout pregnancy and up until weaning. Data are mean ± SEM and analysed by two-way ANOVA with treatment (*P*_trt_) and sex (*P*_sex_) as major factors. *P*_int_ represents the interaction between treatment and sex. Multiple comparisons were determined by Sidak posthoc testing. * = *p* < 0.05, ** = *p* < 0.01 compared to control of same sex. *n* = 6–9.

**Figure 3 nutrients-12-00267-f003:**
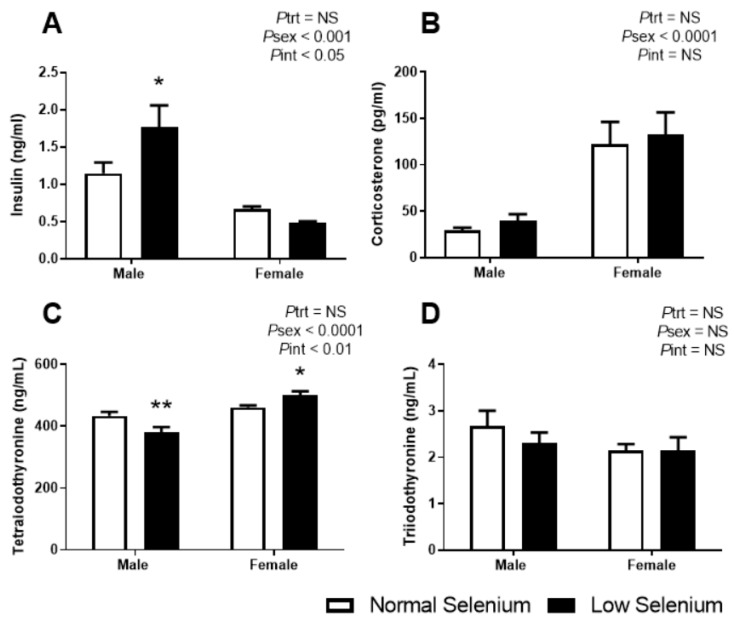
Offspring plasma hormone levels. PN180 plasma levels of (**A**) insulin, (**B**) corticosterone, (**C**) tetraiodothyronine (thyroxine) and (**D**) triiodothyronine. Open bars (white) indicate offspring from litters that consumed a normal selenium diet, whereas closed bars (black) are offspring from litters that consumed a low selenium diet throughout pregnancy and up until weaning. Data are mean ± SEM and analysed by two-way ANOVA with treatment (*P*_trt_) and sex (*P*_sex_) as major factors. *P*_int_ represents the interaction between treatment and sex. Multiple comparisons were determined by Sidak posthoc testing. * = *p* < 0.05, ** = *p* < 0.01 compared to control of same sex. *n* = 6–9.

**Figure 4 nutrients-12-00267-f004:**
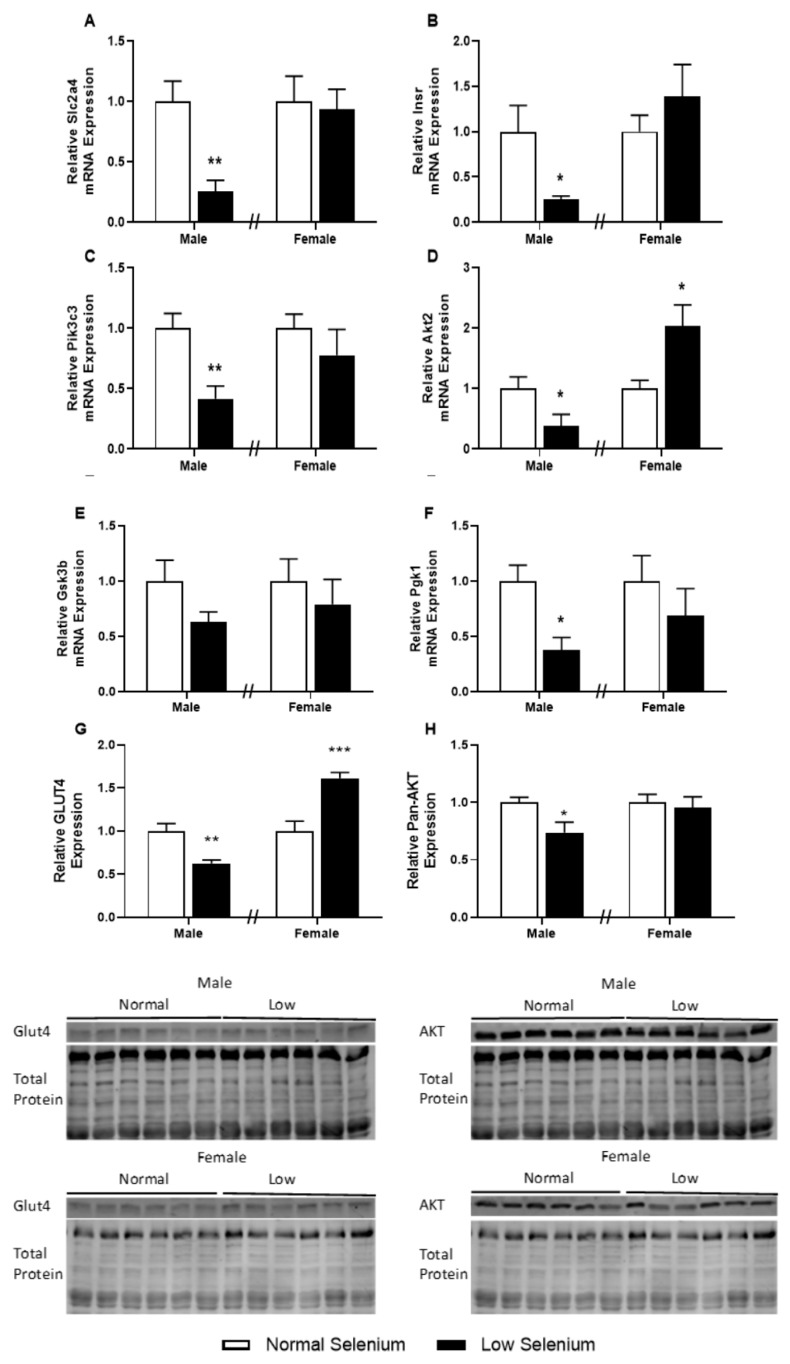
Insulin receptor pathway in the skeletal muscle of offspring. PN180 insulin receptor pathway mRNA and protein expression in the gastrocnemius. Expression of (**A**) *Slc2a4*, (**B**) *Insr*, (**C**) *Pik3c3*, (**D**) *Akt2*, (**E**) *Gsk3b* and (**F**) *Pgk1* in males and females at PN180. Protein expression of (**G**) GLUT4 and (**H**) Pan-AKT is shown in both males and females. Open bars (white) indicate offspring from litters that consumed a normal selenium diet, whereas closed bars (black) are offspring from litters that consumed a low selenium diet throughout pregnancy and up until weaning. Data are mean ± SEM and analysed by unpaired *t*-tests between mice for litters on a normal selenium diet and a low selenium diet in both males and females. * = *p* < 0.05, ** = *p* < 0.01, *** = *p* < 0.001 compared to control of same sex. *n* = 6–8.

**Figure 5 nutrients-12-00267-f005:**
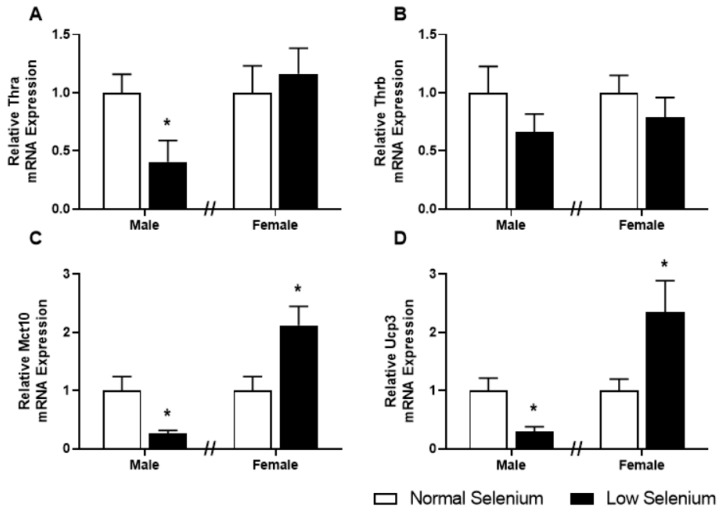
Thyroid hormone transporters and receptors in offspring skeletal muscle. PN180 mRNA expression in the gastrocnemius of thyroid hormone transporters and receptors. Expression of (**A**) *ThrA,* (**B**) *ThrB*, (**C**) *Mct10* and (**D**) *Ucp3* in males and females at PN180. Open bars (white) indicate offspring from litters that consumed a normal selenium diet, whereas closed bars (black) are offspring from litters that consumed a low selenium diet throughout pregnancy and up until weaning. Data are mean ± SEM and were analysed by unpaired t-tests between mice for litters on a normal selenium diet and a low selenium diet in both males and females. * = *p* < 0.05 compared to control of same sex. *n* = 6–8.

**Table 1 nutrients-12-00267-t001:** qPCR primer list.

Group	Gene Name	*Gene Acronym*	Accession Number	Primer Sequence
Housekeepers	Hypoxanthine Phosphoribosyl transferase 1	*Hprt1*	NM_013556	F′ AGGGATTTGAATCACGTTTG R′ TTTACTGGCAACATCAACAG
	Beta-2-Microglobulin	*B2m*	NM_009735	F′ GTATGCTATCCAGAAAACCC R′ CTGAAGGACATATCTGACATC
Insulin Receptor Pathway	Solute Carrier Family 2 Member 4 (*GLUT4*)	*Slc2a4*	NM_009204	F′ CAATGGTTGGGAAGGAAAAG R′ AATGAGTATCTCATAGGAGGC
	Insulin Receptor	*Insr*	NM_010568	F′ AAGACCTTGGTTACCTTCTC R′ GGATTAGTGGCATCTGTTTG
	Phosphatidylinositol 3-Kinase Catalytic Subunit Type 3	*Pik3c3*	NM_181414	F′ CTATGGAGAATTTAGTGGAGAG R′ CTTCATATGTGAGTTGCTTGG
	Thymoma Viral Proto-Oncogene 2	*Akt2*	NM_007434	F′ GAAAGGAGACTGTAAAAAGTGG R′ ATACAGTATCGTCTTGGGTC
	Glycogen Synthase Kinase 3 Beta	*Gsk3b*	NM_019827	F′ CACTCTTCAACTTTACCACTC R′ ATTAGTATCTGAGGCTGCTG
	Phosphoglycerate Kinase 1	*Pgk1*	NM_008828	F′ CTATCATAGGTGGTGGAGAC R′ ACACTAGGTTGACTTAGGAG
	Uncoupling Protein 3	*Ucp3*	NM_009464	F′ CAAGAAATGCCATTGTCAAC R′ GAAGTTGTCAGTAAACAGGTG
Thyroid Transporters and Receptors	Solute Carrier Family 16 Member 10 (Mct10)	*Slc16a10*	NM_001114332	F′ TCCTATTGCAGGGTTACTTC R′ GATCTTTCTTTGCTTCTTGC
	Thyroid Hormone Receptor α	*Thra*	NM_178060	F′ CATGGACTTGGTTCTAGATG R′ CTGTAGCAACATGTATCAGG
	Thyroid Hormone Receptor β	*Thrb*	NM_001113417	F′ GAGACTCTAACTTTGAATGGG R′ CGATCTGAAGACATTAGCAG

**Table 2 nutrients-12-00267-t002:** PN30 allometry.

	Male		Female		*P* _trt_	*P* _sex_	*P* _int_
(mg)	Control	Low	Control	Low			
Liver	717.37 ± 46.21	785.03 ± 24.44	665.82 ± 38.67	714.83 ± 19.39	0.1507	0.1344	0.8147
Adrenal	2.11 ± 0.24	1.96 ± 0.24	2.03 ± 0.11	1.95 ± 0.24	0.5711	0.8387	0.8722
Brain	374.55 ± 10.92	376.12 ± 10.65	380.90 ± 6.79	377.98 ± 8.97	0.9454	0.6793	0.8208
Gastrocnemius	50.87 ± 1.75	57.05 ± 3.11	48.50 ± 2.01	46.36 ± 3.36	0.4315	**0.0160**	0.1123
Tibialis Anterior	16.55 ± 0.70	20.80 ± 1.87 *	14.49 ± 0.61	15.76 ± 0.95	**0.0231**	**0.0047**	0.2017
Soleus	3.64 ± 0.39	3.63 ± 0.24	3.10 ± 0.13	3.01 ± 0.25	0.8641	0.0534	0.8855
EDL	4.19 ± 0.33	4.93 ± 0.26 *	4.33 ± 0.20	3.59 ± 0.15	0.9992	**0.0257**	**0.0076**
Testes	35.17 ± 1.96	33.71 ± 1.63	-	-			
Ovaries	-	-	4.98 ± 0.88	4.01 ± 0.77			

Data are mean ± SEM with all weights in milligrams. All weights are litter averages for each sex. All measurements of adrenals, gastrocnemius, tibialis anterior, soleus, EDL (extensor digitorum longus), testes and ovaries include the total weight from both the left and right side of the body. Data analysed by two-way ANOVA with treatment (*P*_trt_) and sex (*P*_sex_) as major factors. *P*_int_ represents an interaction between treatment and sex. When a major effect was detected, a Sidak posthoc test was performed. Bold text indicates that a major effect reached significance. * = *p* < 0.05 indicated posthoc analysis compared to control of same sex. Unpaired *t*-test was conducted on reproductive organs. *n* = 6–9 litters.

**Table 3 nutrients-12-00267-t003:** PN180 allometry.

	Male		Female		*P* _trt_	*P* _sex_	*P* _int_
(mg)	Control	Low	Control	Low			
Liver	1431.69 ± 55.10	1347.10 ± 54.17	1080.80 ± 53.76	998.97 ± 37.44	0.9797	**<0.0001**	0.1311
Adrenal	2.09 ± 0.29	1.82 ± 0.30	3.59 ± 0.25	3.55 ± 0.57	0.6604	**<0.0001**	0.7542
Brain	425.30 ± 8.64	428.70 ± 4.09	426.46 ± 5.48	443.66 ± 1.52	0.1205	0.2199	0.2918
Gastrocnemius	146.76 ± 3.80	148.15 ± 3.18	115.42 ± 1.85	114.70 ± 1.75	0.9092	**<0.0001**	0.7183
Tibialis Anterior	41.80 ± 1.51	42.58 ± 2.11	35.86 ± 1.22	35.45 ± 0.77	0.3946	**<0.0001**	0.2460
Soleus	8.52 ± 0.35	8.04 ± 0.33	6.98 ± 0.33	6.19 ± 0.33	0.1338	**<0.0001**	0.7474
EDL	9.84 ± 0.63	9.57 ± 0.19	8.38 ± 0.42	7.92 ± 0.22	0.0721	**<0.0001**	0.6543
Testes	95.11 ± 2.80	91.06 ± 1.73	-	-			
Ovaries	-	-	3.47 ± 0.35	4.83 ± 1.28			

Data are mean ± SEM with all weights in milligrams. All weights are litter averages for each sex. All measurements of adrenals, gastrocnemius, tibialis anterior, soleus, EDL (extensor digitorum longus), testes and ovaries include the total weight from both the left and right side of the body. Data analysed by two-way ANOVA with treatment (*P*_trt_) and sex (*P*_sex_) as major factors. *P*_int_ represents an interaction between treatment and sex. When a major effect was detected, a Sidak posthoc test was performed. Bold text indicates that a major effect reached significance. Unpaired *t*-test was conducted on reproductive organs. *n* = 6–8 litters.

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
