# Peer review of "Maternal Selenium Deficiency in Mice Alters Offspring Glucose Metabolism and Thyroid Status in a Sexually Dimorphic Manner"

_nutrients, 2020, doi:10.3390/nu12010267_

Round 1

Reviewer 1 Report

The paper describes an interesting study examing the effects of maternal dietary selenium level on offspring metabolism, including showing impaired glucose tolerance. The paper was well written, though a few more details could be added to the methods.

[Line 18] Does PN mean postnatal day?

[Line 20] What does P trt stand for?

[Line 23] Should P < 0.041 be =? 

[Line 33-34] Do you mean Type 2 diabetes mellitus specifically, or all types of diabetes?

[Lines 34-35] Are the diabetes statistics worldwide, or country specific?

[Lines 44] Tha acronymns GDM and IUGR do not need to be defined if they are not used again.

[Line 88] What is meant by 'littered down'? Gave birth?

[Line 90] Is there a measurable selenium level in chow?

[Line 91] How was consumption monitored? Per cage or individual? By hand or where they placed in an electronically monitored cage?

[Statistical analysis] Was any correction made for litter/litter size/dam?

[Line 303] The acronym DOHaD has already been used, so perhaphs should be defined there.

[Conclusion] Large parts of this section are more 'future studies' than an actual conclusion.

Author Response

We thank you for your careful review and  opportunity to revise our manuscript and address the comments for publication in Nutrients. Please find below a point by point response to each of your comments.

Reviewer 1

[Line 18] Does PN mean postnatal day?

PN does stands for postnatal day. We have gone through the abstract and removed this abbreviation. Similarly we have now avoided using any other abbreviations in the abstract where possible. As this increased the word count to beyond 200 words, we have removed P values from the abstract to allow the words to still fit in the limited space.

[Line 20] What does P trt stand for?

Ptrt stands for the effects of maternal selenium deficiency (Treatment) and is defined in line 164-166. We understand that this may be confusing in the abstract. As such, we have removed this abbreviation. In order to keep the word limit to the 200 words requested for this journal, we have removed P values altogether from the abstract

[Line 23] Should P < 0.041 be =? 

Apologies for this mistake, you are correct. In line with comment 1 an 2, we have removed this error.  

[Line 33-34] Do you mean Type 2 diabetes mellitus specifically, or all types of diabetes?

We acknowledge this needs to be more specific. In lines 32-34 we mean T2DM. In lines 34-37 we mean diabetes mellitus (1 and 2). We have now gone through the manuscript and corrected our terminology accordingly.

[Lines 34-35] Are the diabetes statistics worldwide, or country specific?

These are worldwide statistics. We have now edited this line as to acknowledge that the statistics are worldwide.

[Lines 44] The acronyms GDM and IUGR do not need to be defined if they are not used again.

Apologies for the insertion of these acronyms. They have subsequently been removed from line 44.

[Line 88] What is meant by 'littered down'? Gave birth?

Yes. This line has been changed from “littered down” to “gave birth” to avoid any future confusion.

[Line 90] Is there a measurable selenium level in chow?

Yes, the quantity of selenium within the normal animal chow was 0.23 mg/kg (230 µg/kg). The animal chow was Teklad Global 18% Protein Rodent Diet Irradiated, ENVIGO, Madison, WI, USA. This additional information has now been added to Line 90 in the methods section.

[Line 91] How was consumption monitored? Per cage or individual? By hand or where they placed in an electronically monitored cage?

Food and water consumption was measured by manually weighing food and water each day and calculating the difference between days. This was done per cage which contained 2 or 3 mice. Approximate values representative  of individual values were calculated by dividing by the number of mice present. This has now been clarified within the methods.

[Statistical analysis] Was any correction made for litter/litter size/dam?

No corrections were made to litter size as litter size was equal between treatment groups. Furthermore, only one male and one female per litter was used for each experimental endpoint to avoid bias induced by having multiple animals used from one litter. While litter size was not impacted by treatment, there was a slight sex bias towards males in the LS group. This was driven by some LS litters having only males or only one female. This contributed to the fact that only n=6 females were available despite there being 8 litters to select from. We have added details regarding this to the statistics and now included litter size and number of males and females per litter in the results sections were appropriate.

[Line 303] The acronym DOHaD has already been used, so perhaps should be defined there.

The acronym “DOHaD” has now been defined in the methods section (line 89) and the definition has been subsequently removed from the discussion.

[Conclusion] Large parts of this section are more 'future studies' than an actual conclusion.

We agree with the reviewer that the conclusion section does have lots of references towards “future studies”. We have restructured the conclusion so that it no longer contains any recommendations to future studies and placed these recommendations in the discussion (Lines 408-414).

Reviewer 2 Report

Refers to the manuscript: Maternal selenium deficiency in mice alters offspring glucose metabolism and thyroid status in a sexually dimorphic manner.

The aim of the study was to assess the effect of selenium defficiency during pregnancy on offspring metabolic functions.

Congratulations to the authors.

The goal has been achieved.

The study and results are interesting. Relatively to the purpose of the study, the authors discussed glucose and thyroid metabolism and differences by sex offspring.

Because it is also important to emphasize the clinical significance of the results of this selenium test in a pregnant women, I suggest supplementing the Introduction or Discussion with a few examples of clinical trials (the number of these trials is very limited).

Pregnancy induced hypertension-PIH (including preeclampsia and gestational hypertension) develops de novo after the 20th week of gestation and disappears by the 12th week after delivery. But distant effects include metabolic disorders in the future life of the mother and child. IUGR is a special case, but small-for-gestational age birth weight is also associated with "programming" metabolic disorders in future life.Nutrients 2019, 11, 1028;  Br.J.Nutr. 2015, 113, 249-258;  Nutrients 2019, 11, 2298;  Nutr. Res. 2006, 26, 497-502

Kind regards,

Author Response

We thank the reviewer for their kind comments regarding our manuscript. Please see below the response the main comment provided.

Because it is also important to emphasize the clinical significance of the results of this selenium test in a pregnant woman, I suggest supplementing the Introduction or Discussion with a few examples of clinical trials (the number of these trials is very limited).

We acknowledge the need for additional references to the clinical studies previously performed and have now included additional references in the introduction and discussion. The references you have suggested have been included in these changes.